# Translational Research in Retinopathy of Prematurity: From Bedside to Bench and Back Again

**DOI:** 10.3390/jcm10020331

**Published:** 2021-01-18

**Authors:** Mitsuru Arima, Yuya Fujii, Koh-Hei Sonoda

**Affiliations:** 1Department of Ophthalmology, Graduate School of Medical Sciences, Kyushu University, Fukuoka 8128582, Japan; aceinter0426@yahoo.co.jp (Y.F.); sonodak@med.kyushu-u.ac.jp (K.-H.S.); 2Center for Clinical and Translational Research, Kyushu University Hospital, 3-1-1 Maidashi, Higashi-ku, Fukuoka 8128582, Japan

**Keywords:** retinopathy of prematurity, oxygen-induced retinopathy, clinical trial, translational research

## Abstract

Retinopathy of prematurity (ROP), a vascular proliferative disease affecting preterm infants, is a leading cause of childhood blindness. Various studies have investigated the pathogenesis of ROP. Clinical experience indicates that oxygen levels are strongly correlated with ROP development, which led to the development of oxygen-induced retinopathy (OIR) as an animal model of ROP. OIR has been used extensively to investigate the molecular mechanisms underlying ROP and to evaluate the efficacy of new drug candidates. Large clinical trials have demonstrated the efficacy of anti-vascular endothelial growth factor (VEGF) agents to treat ROP, and anti-VEGF therapy is presently becoming the first-line treatment worldwide. Anti-VEGF therapy has advantages over conventional treatments, including being minimally invasive with a low risk of refractive error. However, long-term safety concerns and the risk of late recurrence limit this treatment. There is an unmet medical need for novel ROP therapies, which need to be addressed by safe and minimally invasive therapies. The recent progress in biotechnology has contributed greatly to translational research. In this review, we outline how basic ROP research has evolved with clinical experience and the subsequent emergence of new drugs. We discuss previous and ongoing trials and present the candidate molecules expected to become novel targets.

## 1. Introduction

### 1.1. Background

Retinopathy of prematurity (ROP) is a representative vision-threatening disease in childhood that is characterized by abnormal retinal vessel growth at the boundary of the vascularized and avascular peripheral retinas of preterm infants [1,2]. Blencowe et al. reported that severe visual impairment, including blindness, and moderate visual impairment caused by ROP affect approximately 20,000 and 12,300 people, respectively, per annum [3]. Improved management of preterm infants, including respiratory support and postnatal nutrition intervention, has gradually reduced the incidence of ROP [4]. However, this improved maternal and neonatal management has resulted in an increase in the survival rate of extremely preterm infants with a high risk of developing ROP [5]. Younger gestational age (GA) and lower birth weight (BW) are the most well-known risk factors associated with the development and severity of ROP [6], and major clinical studies have demonstrated a gradual decline in GA and BW [7]. Thus, the proportion of patients with severe ROP requiring treatment may increase in the future. The retina of preterm infants is still immature [8,9] and retinal vascularization and neurodevelopment interfere with each other [10]. Clinical studies using optical coherence tomography (OCT), OCT angiography, and electroretinogram techniques have shown that the more severe the ROP, the more impaired the development of the macular structure, foveal vascular formation, and synapse formation between retinal neurons [11,12,13,14]. Therefore, preventing the progression of ROP is very important to protect the lifelong quality of vision of preterm infants.

### 1.2. Unmet Medical Needs for ROP Treatment

Extremely preterm infants are extremely vulnerable to external invasion [15,16,17]. In infants with poor general health conditions, stress due to medical examination and treatment can lead to circulatory and respiratory failure [18,19,20]. Hence, the development of less invasive treatments (progression in prophylactic and therapeutic agents) is urgently required.

Advances in the comprehensive analysis of genes and proteins have made it possible to identify specific molecules involved in the pathogenesis of diseases [21,22,23]. “Molecular-targeted therapy” that inhibits or promotes the action of these specific molecules was developed mainly in the field of oncology [24,25,26], but has also been applied in the field of ophthalmology [27,28]. Intravitreal injection of anti-vascular endothelial growth factor (VEGF) agents (anti-VEGF therapy) is the most common molecular-targeted therapy in ophthalmology and is commonly used for various retinal diseases, such as age-related macular degeneration, diabetic retinopathy, and retinal vein occlusion [29,30,31]. Unfortunately, anti-VEGF therapy for ROP is not yet optimal. Anti-VEGF therapy can develop residual non-vascularized areas because of the inhibition of normal retinal vascularization, leading to very late recurrence [32,33,34,35]. There is no consensus regarding the long-term safety of anti-VEGF treatment, and some researchers are concerned that the long-term suppression of systemic VEGF expression may cause developmental delays [36,37,38,39,40,41]. Therefore, there is an urgent need to develop new therapies “together with or beyond anti-VEGF therapy.”

Although various studies have been conducted from a clinical perspective to prevent blindness caused by ROP, including the identification of risk factors, development of predictive models, and diagnosis of the disease stage with artificial intelligence [42,43,44,45,46,47,48,49,50,51], this review will focus on the translational research that has progressed from basic research to clinical trials.

## 2. Relationship between Oxygen Concentration and Retinal Neovascularization

### 2.1. The First ROP Epidemic: Learning from the Bedside

In the early 20th century, oxygen administration was proven to be effective in the treatment of apnea and cyanosis in preterm infants; therefore, long-term oxygen administration was used for the treatment of these conditions worldwide [52]. However, this unmonitored and unrestricted high oxygen supplementation resulted in a high incidence of ROP in the 1940s and 1950s (the first ROP epidemic) [2,53]. ROP was first described as retrolental fibroplasia (RLF) by Terry in 1942 [54], who reported the detailed ophthalmoscopic characteristics of RLF [55]; Owens et al. and Reese et al. created the RLF staging system [56,57]. These invaluable findings have contributed to the establishment of the current international classification of ROP [58,59]. The ROP classification is summarized in Table 1. Szewczyk et al. reported that vascular dilatation, tortuosity, and abnormal neovascularization occurred after withdrawal from hyperoxia [60]. They also demonstrated that re-exposing patients to hyperoxia reduced the activity of RLF [60]. In 1950s, there were various theories regarding whether RLF occurs because of anoxia, oxygen toxicity (high oxygen concentration), or relative hypoxia caused by oxygen withdrawal [61,62,63]. Although the specific mechanism was inconclusive at that time, it was gradually accepted that oxygen overuse was involved in the progression of ROP, because limited oxygen supplementation was found to reduce the incidence of severe RLF in the clinical trials conducted by Patzs et al. and Lanman et al. [64,65].

### 2.2. Advances in Respiratory Management: Reduction of Oxygen Supplementation

The recognition of RLF as an iatrogenic disease caused by oxygen overuse led to the initiation of oxygen monitoring studies [66,67,68,69]. In addition, the development of non-invasive and continuous monitoring systems for blood oxygen saturation accelerated our understanding of the role of oxygen in ROP pathogenesis [70,71,72]. Flynn et al. explored the threshold level of oxygen saturation using a transcutaneous partial pressure oxygen (tcPO_2_) monitor and revealed that a tcPO_2_ level of ≥80 mmHg for ≥12 h a day doubled the risk of severe ROP [73]. Thereafter, many randomized clinical trials (the Supplemental Therapeutic Oxygen for Prethreshold Retinopathy of Prematurity (STOP-ROP) trial; the Surfactant, Positive Airway Pressure, Pulse Oximetry Randomized trial; the Canadian Oxygen Trial; and the Benefits of Oxygen Saturation Targeting Study (BOOST) and BOOST-II trials) were conducted using pulse oximetry, which is widely used today [74,75,76,77,78,79]. The results of these trials indicated that lower oxygen levels reduced the incidence of ROP, but in the BOOST-II trial, the mortality rate was higher in the lower saturation of the percutaneous oxygen (SpO_2_) target group (85–89%) than in the higher SpO_2_ target group (91–95%) [74,75,76,77,78,79]. A meta-analysis of patient data obtained from these five trials also revealed a significantly higher mortality rate in the lower SpO_2_ target group (85–89%) than in the higher SpO_2_ target group (91–95%) [80]. Although the optimal target SpO_2_ for preterm infants remains unknown, both the American Academy of Pediatrics and European consensus guidelines recommend a target SpO_2_ in the range of 90–95% [81,82].

In addition, the widespread use of continuous airway pressure and the administration of caffeine and surfactants to achieve reductions in ventilation- and oxygen-mediated pulmonary injury have reduced the duration of intubation and oxygen use [83,84,85]. These advances in respiratory support have also contributed to a reduction in the incidence of ROP [86].

### 2.3. Establishment of the Animal Model of ROP: Learning from the Bench

The discovery of the involvement of oxygen supplementation in the development of ROP motivated many researchers to examine the effect of oxygen on retinal angiogenesis in neonatal animals [87,88]. One of the first animals to be used for the study of ROP was the kitten. Ashton et al. reported that vaso-obliteration of the newly formed capillaries occurred under hyperoxia and that vaso-proliferation appeared a few days after returning from hyperoxia to ambient air [89]. This was a highly valuable discovery that contributed to the establishment of the two-phase theory of ROP (Figure 1). The oxygen-induced retinopathy (OIR) phenotype in kittens replicated human ROP and became a popular and suitable model for studying the pathogenesis of ROP. Ashton et al. also described that swelling of the retinal tissue around the vessels was the primary factor concerning the promotion of vaso-obliteration [90]. Similar pathological features were observed in the autopsied eyes of ROP patients, and similar capillary obliteration was also observed on the study of ROP patients using fluorescein angiography, which demonstrated the efficacy of OIR in elucidating the pathogenesis of ROP. Dollery et al. found that administration of high concentration oxygen leads to vaso-constriction in the human retina [91]. They also experimentally demonstrated that oxygen uptake from the retinal vessels appeared to decrease, and that all retinal layers, not only the deep layer, received oxygen only from the choroidal blood flow when the partial pressure of arterial oxygen (PaO_2_) increased [92]. These results supported the hypothesis that withdrawal from oxygen administration leads to relative retinal hypoxia, which promotes pathological angiogenesis in preterm infants. Abnormal vascular network formation in the developing retina inhibits the proper development of the neural retina. In addition, fluctuations in oxygen levels (both hyperoxia and hypoxia) are also known to promote neural damage [93,94]. Similar to humans, disruption of the retinal layer structure and reduction of synapse formation have been reported in OIR [95]. Based on the evidence from multiple basic and clinical studies, it is currently recognized that the progression of ROP consists of the following two phases: in phase 1, wherein hyperoxia causes vaso-obliteration and cessation of vascular growth; and in phase 2, wherein withdrawal of oxygen administration causes the retina to fall into relative hypoxia, which subsequently promotes vaso-proliferation (Figure 1A) [96].

Since the vascular development of kittens and puppies is similar to that of humans, many studies were conducted to elucidate the mechanism of onset and severity of ROP using OIR generated in these animals by continuous oxygen loading (70–80% O_2_ for 4 days at least in kittens and 95–100% O_2_ for 4 days in puppies) [89,97,98,99,100,101]. The puppy OIR differs from the kitten OIR in that it can reproduce the retinal detachment observed in the late severe stage of human ROP (Stage 4 and 5, Table 1) [102]. OIR was also created in rabbits [89]. However, in rabbits, the retinal vessels were limited to a small region of the retina and did not mimic the retinal vascular development found in the human. Most studies are currently conducted in rodents such as mice or rats [103,104,105]. The advantages of using rodents are the following: (1) genetic manipulation is easier, (2) reagents such as recombinant proteins and antibodies are readily available, and (3) the cost of raising them is lower than that for large animals. The rat OIR was first reported by Patzs et al. [106]. However, it was difficult to reproduce the vaso-proliferative response [107], and therefore the optimal oxygen loading concentration was investigated by many researchers. Penn et al. reported that peripheral retinal neovascularization did not occur in the group continuously exposed to 80% O_2_ for 2 weeks, whereas it occurred in the 80%/40% group (a cycle of 12 h exposure to 80% O_2_ followed by 12 h exposure to 40% O_2_ for 2 weeks) [108], indicating that fluctuations in oxygen concentration play a crucial role in abnormal retinal angiogenesis. Penn et al. further verified that the 50%/10% group (a cycle of 24 h exposure to 50% O_2_ followed by 24 h exposure to 10 % O_2_ for 2 weeks) produced more peripheral retinal neovascularization than the 80%/40% group (a cycle of 24-h exposure to 80% O_2_ followed by 24-h exposure to 40% O_2_ for 2 weeks) [103]. The mouse OIR was established by Smith et al. [104]. In this model, exposure to continuous hyperoxia (constant exposure of 75% O_2_) for 5 days from postnatal day (P) 7 causes vaso-obliteration, and relative hypoxia due to withdrawal from hyperoxia induces vaso-proliferation [105]. In mice, continuous hyperoxic exposure can induce adequate retinal neovascular growth. Recently, in addition to the model developed by Smith et al. [104], a model in which mice are exposed to 85% O_2_ for 3 days from P8 has been used [109].

In this section, the pathogenesis of ROP was described, focusing on the relationship between oxygen levels and angiogenesis. It was found that relative hypoxia triggers vaso-proliferation, which led to the development of therapies to ablate the non-vascularized retina (Section 3.1 and Section 3.2). In addition, detailed analysis of OIR retinas has revealed that not only VEGF (Section 3.3) but also various other factors such as other growth factors, hematopoietic hormones, reactive oxygen species, and inflammation (Section 4 and Section 5) are involved in the pathogenesis of ROP. In this review, we will focus on existing therapies and therapies of which efficacy have been investigated in clinical trials. We would like to show how each target molecule is involved in the development and severity of ROP.

## 3. Existing Treatments for ROP

### 3.1. Cryotherapy

Trans-scleral cryotherapy has been used to treat ROP in Japan since 1972, and its use has gradually gained traction around the world [110,111]. The therapeutic efficacy of cryotherapy was later established by the Multicenter Trial of Cryotherapy for Retinopathy of Prematurity (CRYO-ROP) in 1988 [112]. This trial demonstrated that cryotherapy significantly reduced the progression of threshold ROP (among severe ROP patients, 50% of the untreated patients are at risk of developing retinal detachment) [112]. However, a 10 year evaluation of the patients enrolled in the CRYO-ROP showed that 44.4% of the patients treated with cryotherapy had visual acuity of 20/200 or less [113]. This led to a reconsideration of the stage of ROP at which therapeutic intervention should be initiated. Therefore, a clinical trial to investigate the efficacy of early intervention for ROP (the Early Treatment for ROP (ETROP) trial) was conducted [114].

### 3.2. Laser Photocoagulation

Cryotherapy has limited efficacy because it was difficult to coagulate the large avascular area extending to the posterior retina (i.e., zone I ROP) [115]. Laser photocoagulation by an indirect ophthalmoscope was expected to overcome this limitation of cryotherapy; therefore, in the 1990s, a number of trials compared the therapeutic effects of laser photocoagulation and cryotherapy [116,117,118,119]. After confirmation that laser photocoagulation was as effective as cryotherapy and offered a better visual prognosis than cryotherapy, laser photocoagulation gradually supplanted cryotherapy [120,121,122]. The final results of the ETROP trial established the efficacy of early intervention with laser photocoagulation in 2004 [123], and laser photocoagulation has since been used as the first-line therapy worldwide. The ETROP study group also indicated that laser photocoagulation should be considered for the eye with type I ROP (i.e., zone I, any stage ROP with plus disease; zone I, stage 3 ROP, with or without plus disease; or zone II, stage 2 or 3 ROP, with plus disease) [123].

Both cryotherapy and laser photocoagulation are based on the concept of eliminating retinal hypoxia by ablation of the avascular area. Originally, patients with ROP tended to develop myopia because of the arrested development of the anterior segment [124,125], and retinal ablative therapy further exacerbates these ocular structural changes [126]. Scar formation promotes a steeper corneal curvature, greater lens thickness, and shallower anterior chamber depth, thereby aggravating myopic changes [126,127]. In addition, since laser photocoagulation is a highly invasive treatment, the general condition of patients may worsen after treatment. Anti-VEGF therapy is valuable in that it overcomes these two major problems.

### 3.3. Anti-VEGF Therapy

#### 3.3.1. Bioactivity of VEGF

In 1983, Senger et al. identified a protein that causes vascular leakage in cancer and named it vascular permeability factor (VPF) [128]. In 1989, Ferrara et al. found a protein that promoted mitogen and the proliferation of vascular endothelial cells and named this factor VEGF [129]. Thus, VPF and VEGF were isolated independently, but sequencing revealed that they are the same molecule [130,131].

The upregulation of VEGF in various retinal diseases including ROP is known to promote pathological angiogenesis [132,133]. Ikeda et al. reported that hypoxia in local tissues is an essential factor in enhancing VEGF production [134]. They further showed that the binding of hypoxia-inducible factor-1 (HIF-1) to the enhancer region of VEGF is essential for hypoxia-stimulated VEGF transcriptional enhancement [135]. HIF-1 is a transcription factor that functions as a master regulator of oxygen homeostasis and is composed of two different subunits: HIF-1α and β [136]. Under normoxia, HIF-1α is hydroxylated by prolyl-hydroxylase (PHD) and is rapidly degraded via the ubiquitin-proteasome system. However, under hypoxia, the expression of HIF-1α is stabilized because of the lack of oxygen, a substrate for PHD. HIF-1α forms a dimer with HIF-1β and enhances the transcription of target genes, including VEGF [137,138]; that is, the enhancement of VEGF transcription by HIF-1 can be considered as a reasonable molecular mechanism that immediately responds to retinal hypoxia and causes angiogenesis to improve ischemia. However, as described below (see Section 3.3.2.), if VEGF expression is increased too much, pathological retinal neovascularization can occur.

The VEGF family comprises VEGF-A, VEGF-B, VEGF-C, VEGF-D, and placental growth factor (PlGF), while the VEGF receptors (VEGF-Rs) consist of three protein-tyrosine kinases: VEGF-R1, VEGF-R2, and VEGF-R3 [139]. Among the VEGF family, VEGF-A plays a key role in angiogenesis [140]. Although VEGF-A can bind to VEGF-R1 and VEGF-R2, angiogenesis is mainly controlled by the VEGF/VEGF-R2 pathway [141]. VEGF-R1 is expressed on the cell membranes of not only vascular endothelial cells, but also of monocytes and macrophages. The VEGF/VEGF-R1 pathway promotes the migration of monocytes and macrophages and production of angiogenic cytokines and chemokines from monocytes and macrophages [141]. The VEGF/VEGF-R signaling pathways involved in angiogenesis are summarized in Figure 2.

#### 3.3.2. VEGF Expression in Phase 1 and Phase 2

Gene knockout studies in mice have demonstrated that VEGF (VEGF-A) is an essential molecule not only for pathological angiogenesis, but also for physiological angiogenesis [142]. In preterm infants, the peripheral retina is not fully vascularized, and thus the blood vessels need to grow properly with the assistance of VEGF after birth. However, in some preterm infants, vascular growth is arrested because of reduced VEGF expression (phase 1).

After birth, the preterm infant is abruptly exposed to relative hyperoxia. Even in the atmosphere, the oxygen concentration is higher than in the intrauterine environment [143]. In the presence of respiratory or cardiovascular disease, oxygen administration further increases the blood oxygen saturation. Suppression of VEGF expression by hyperoxia halts normal retinal vascular development and leads to apoptosis of immature vascular endothelial cells, thus completing the extensive avascular region [144,145]. The improved metabolic function with the growth of preterm infants promotes VEGF production in the ischemic retina [96,146]. The excessive production of VEGF in response to hypoxia leads to pathological vascular proliferation. These abnormal vessels can form proliferative membranes and cause tractional retinal detachment, resulting in irreversible visual impairment (phase 2).

#### 3.3.3. Clinical Trials with Anti-VEGF Therapy

Since the inhibition of VEGF in phase 2 inhibited pathological angiogenesis in vivo [146,147], several pilot studies were conducted to assess the efficacy of intravitreal injection of bevacizumab (IVB) for severe ROP [148,149,150]. Because retinal detachment was accelerated after treatment in some stage 4 and stage 5 patients [151], a large clinical trial (the Bevacizumab Eliminates the Angiogenic Threat of ROP (BEAT-ROP) trial) was conducted to determine the efficacy of bevacizumab in stage 3 ROP [152]. The trial demonstrated that the recurrence rate was lower after IVB than after laser photocoagulation for zone I ROP [152]. Another advantage of IVB is that it is less invasive for preterm infants because it requires less treatment time than laser photocoagulation. Therefore, the off-label use of bevacizumab has come to be a first-line treatment for preterm infants with severe ROP or an unstable general condition [153,154]. The off-label use of bevacizumab limited the number of centers at which anti-VEGF therapy can be performed. However, a study that investigated ranibizumab therapy for the treatment of infants born prematurely with ROP demonstrated that intravitreal injection of ranibizumab (IVR) was as effective as laser photocoagulation [155]; it is, thus, expected that the frequency of anti-VEGF therapy will increase in the future. Unlike laser treatment, anti-VEGF therapy does not cause ocular structural sequelae, and therefore is unlikely to induce high myopia [153,156]. The reduced risk of amblyopia is another benefit of anti-VEGF therapy.

However, anti-VEGF therapy has some limitations. Anti-VEGF agents also interfere with the normal development of the retinal vascular network, resulting in a long-lasting residual avascular area in the peripheral retina [32]. Residual ischemia is a risk factor for ROP recurrence, making it difficult for us to determine whether to extend the interval between patient visits. Although vitrectomy may be necessary for cases of ROP recurrence [157,158], few centers can perform vitrectomy for infants. Some physicians perform the additional laser treatment to prevent late recurrence in cases of residual avascular areas several weeks or months after anti-VEGF therapy. There are also concerns that anti-VEGF therapy, particularly IVB, may cause neurodevelopmental delay [39,40,159]. Although no consensus has been established because some reports indicate that IVB does not affect neurodevelopment [38,160,161], it is known that serum VEGF levels are significantly reduced for approximately 2 months after IVB [36,37]. VEGF is an essential molecule in brain homeostasis, as well as angiogenesis [162,163]. Thus, the possibility cannot be denied that prolonged VEGF suppression results in the inhibition of proper neurodevelopment. In contrast, IVR does not reduce the serum VEGF level [155]. If prolonged serum VEGF suppression affects neurodevelopment, IVR may be safer than IVB. However, IVR has the disadvantage that it is associated with a higher recurrence rate (30–50%) than IVB (5–25%) [155,164,165].

The Aflibercept for ROP-Intravitreal Injection Versus Laser Therapy trial is currently underway. Aflibercept can bind not only to VEGF-A, but also to VEGF-B and PlGF [166]. In contrast to VEGF-A, neither VEGF-B nor PlGF is essential for physiological vascularization. However, PlGF and VEGF-B, as with VEGF-A, are involved in pathological vascularization [167,168]. Hence, aflibercept is the most potent inhibitor of pathological angiogenesis among anti-VEGF agents. Indeed, it was reported that the recurrence rate of ROP was lower in the group treated with intravitreal injection of aflibercept (IVA) than in the group treated with IVR [169]. However, it was also reported that the serum VEGF level was significantly reduced for 3 months after IVA [170]. A report on the efficacy and safety results of IVA is awaited.

The targets of the anti-VEGF agents bevacizumab, ranibizumab, and aflibercept are summarized in Figure 2.

## 4. Translational ROP Research: Previous and Ongoing Clinical Trials

### 4.1. Antioxidants

Owen et al. demonstrated the efficacy of vitamin E in the treatment of ROP in 1949 [171]. Several clinical studies investigating vitamin E administration have since been performed. Although the results were not consistent, with some reports finding an effect and others not [172,173], two meta-analyses showed that vitamin E supplementation significantly reduced the severity of ROP [174,175]. However, the systemic administration of vitamin E has the disadvantage of increasing the incidence of sepsis and necrotizing enterocolitis [176,177]. Preterm infants are immediately exposed to hyperoxic conditions after birth [96]. This oxidative stress induces the production of reactive oxygen species (ROS). ROS are intimately involved in the pathogenesis of ROP. ROS contribute to both vascular endothelial cell apoptosis under hyperoxia (phase 1) and pathological angiogenesis under relative hypoxia (phase 2) [178,179,180]. ROS also promote VEGF upregulation via the HIF-1/VEGF signaling pathway [181]. It is well-known that the biological activity of antioxidant enzymes in preterm infants is lower than that in full-term infants [182,183]. Because vitamin E is a fat-soluble antioxidant that can scavenge ROS [184], it is considered that vitamin E administration can inhibit ROP progression (Figure 1B).

In addition to vitamin E, many other clinical trials have been conducted to evaluate the effects of antioxidant treatments. Soghier et al. performed a randomized clinical trial to evaluate the effect of the nicotinamide adenine dinucleotide phosphate inhibitor, *n*-acetyl cysteine, in reducing ROP severity, but no effect was found [185]. Manzoni et al. assessed the efficacy of carotenoid (lutein and zeaxanthin) supplementation, however a significant effect was not obtained [186]. Since the anti-rheumatic drug, D-penicillamine, is also known to have antioxidant properties [187], several trials were conducted to test its effects [188,189]; however, compared with controls, patients treated with D-penicillamine showed no significant effect [188,189]. Parad et al. performed a multicenter trial to investigate the effects of superoxide dismutase (SOD) and found that SOD reduces the risk of ROP development [190]. Coenzyme Q10 (CoQ10) is also a known antioxidant that directly scavenges free radicals and indirectly regenerates vitamin E. Beharry et al. showed that CoQ10 supplementation reduced the severity of OIR [191]. Akdogan et al. retrospectively investigated the efficacy of eye drops containing CoQ10 and vitamin E. They reported that this treatment induced faster retinal full vascularization in infants with stage 0 and 1 ROP and reduced the number of laser photocoagulation treatments, which were performed as a rescue treatment, in infants with aggressive posterior ROP [192].

These results suggest that the antioxidants have the potential to become a new treatment modality for ROP. Meta-analyses have demonstrated that consumption of human milk reduces the incidence of severe ROP [4], which is thought to be mediated by the antioxidants present in human milk (Figure 1) [193,194].

### 4.2. Vitamin A

Vitamin A is a fat-soluble micronutrient essential for normal visual function [195]. In preterm infants, blood levels of vitamin A and its metabolites are very low because the maternal supply of vitamin A is interrupted and the hepatic storage capacity of vitamin A is immature [196,197]. Several clinical trials have indicated that vitamin A administration reduced the incidence and severity of ROP [198,199]. The effects of vitamin A can be mediated by two main possible mechanisms.

Vitamin A is necessary for lung maturation and promotes epithelial cell differentiation and proliferation, as well as the secretion of surfactants [200]. There is strong evidence for an association between vitamin A administration and a reduction in chronic lung disease and bronchopulmonary dysplasia in preterm infants [201,202,203,204]. Shortening the duration of oxygen administration by reducing the prevalence of lung disease may reduce the development and severity of ROP.

In addition, although it was previously known that vitamin A has an inhibitory effect on tumor angiogenesis [205,206], Ozkan et al. first showed that vitamin A treatment inhibited hypoxia-induced retinal neovascularization via the suppression of VEGF production using OIR rats [207]. Pal et al. demonstrated that retinoic acid, a metabolite of vitamin A, led to the suppression of vascular barrier dysfunction and endothelial migration via the inhibition of VEGF/VEGF-R2-induced activation of phospholipase C-γ and increase in cGMP (Figure 1B) [208].

Thus, vitamin A inhibits ROP progression through both direct and indirect mechanisms. Meta-analyses have also shown the efficacy of vitamin A [4] and it is now considered a potential novel therapeutic strategy.

### 4.3. Adrenocorticotropic Hormone and Steroids

In 1951, Reese et al. reported the possibility of inhibiting ROP progression using adrenocorticotropic hormone (ACTH) [57]. However, the trials conducted the following year failed to yield reproducible results, and ACTH treatment gradually ceased to be performed after the early 1950s [209]. ACTH stimulates the production of corticosteroids, which are necessary for fetal maturation [210]. The adrenal glands have markedly different endocrine characteristics in neonates and adults. The zona fasciculata, which produce corticosteroids, are immature in infancy, even in full-term neonates [211]. In addition, preterm infants may exhibit less corticosteroid secretion in response to corticotropin releasing hormone and ACTH stimulation because of the immaturity of the hypothalamic–pituitary–adrenal axis [212,213]; that is, preterm infants are extremely vulnerable to circulatory failure caused by stress-induced relative adrenal insufficiency [214]. We have reported that the such development of circulatory failure is associated with the development and severity of ROP [20]. There are several possible etiologies of ROP other than circulatory-collapse-induced retinal ischemia. Nuclear transfer of steroids complexed with glucocorticoid receptors (GRs) induces the production of anti-inflammatory proteins such as inhibitor-kappa B (IκB) via binding to the glucocorticoid response element [215]. In addition, the steroid-GR complex acts to inhibit the production of angiogenetic inflammatory cytokines via binding to AP-1 [216,217]. Inadequate corticosteroid levels may attenuate these anti-inflammatory effects and lead to ROP progression. Indeed, hypercytokinemia was found to be more prolonged after the induction of sepsis in a mouse model of relative adrenal insufficiency than in control mice [218].

Several trials demonstrated that antenatal corticosteroid therapy reduced the incidence of ROP [210,219,220]. Two meta-analyses showed that the incidence of ROP was reduced by the early (<8 days) administration of systemic postnatal corticosteroids but increased by the late (>7 days) administration of systemic postnatal corticosteroids [221,222]. The reduced risk of ROP due to the early administration of steroids, such as in those with pulmonary disease and intraventricular hemorrhage, may contribute to the reduction in the development of ROP [221,223].

### 4.4. Erythropoietin

Erythropoietin (EPO) is a hormone that is produced in response to tissue hypoxia and promotes hematopoiesis [224]. It has been reported that EPO expression is also increased in the vitreous fluid of those with ischemic retinal disease [225]. First, HIF-1 is a molecule first identified by Semenza et al. in 1992 as “a factor that induces EPO in a hypoxia-dependent manner,” and EPO, as with VEGF, is transcriptionally activated by HIF [226]. EPO is known to inhibit the apoptosis of ganglion cells and photoreceptor cells and promote angiogenesis in the retina [227,228,229]. Jing et al. showed that EPO exerts a biphasic effect in ROP using OIR mice [230]. Under hyperoxia (phase 1), EPO inhibited vaso-obliteration and reduced the avascular area, whereas under relative hypoxia (phase 2), EPO exacerbated pathological angiogenesis. In addition, the administration of EPO in phase 1 inhibited retinal neuronal apoptosis [230]. This group also reported that the inhibition of EPO in phase 2 inhibited pathological angiogenesis [231]. Yang et al. demonstrated that VEGF-A/VEGF-R2 signaling activates the EPO receptor (EPOR) and that the interaction between phosphorylated EPOR (*p*-EPOR) and *p*-VEGF-R2 promotes retinal pathological angiogenesis [232].

Many trials have investigated the systemic administration of EPO in patients with ROP. Brown et al., Suk et al., and Romagnoli et al. reported that EPO administration increased the risk of ROP [233,234,235]. Randomized clinical trials showed no significant effect of EPO on ROP development [236,237]. Because of the biphasic nature of EPO as described above, studies have evaluated the effects of early EPO administration. However, the expected results have not been obtained [238,239]. A meta-analysis also concluded that EPO administration had no effect on ROP development [4].

### 4.5. Insulin-Like Growth Factor 1 and Insulin-Like Growth Factor Binding Protein 3

Because insulin-like growth factor 1 (IGF-1) levels in the serum and vitreous fluid are associated with the severity of retinopathy in diabetic retinopathy, Smith et al. used OIR to analyze IGF-1 function and demonstrated that IGF-1 is required for pathological angiogenesis in ischemic retinal disease [240]. They further showed that the IGF-1/IGF-1 receptor pathway contributes to the proliferation of vascular endothelial cells by promoting the phosphorylation of MARK, which is downstream of the VEGF/VEGF-R2 pathway [241]. Using IGF-1-deficient mice, Hellström et al. showed that IGF-1 is also essential for physiological angiogenesis; furthermore, they demonstrated that low IGF-1 expression (not exceeding a threshold) does not cause the phosphorylation of Akt, which is necessary for the survival of vascular endothelial cells, even in the presence of VEGF [242]. In addition, they proposed the following pathological hypothesis based on basic experimental results and patient serum data, which is now generally accepted [242]. First, the reduced serum concentration of IGF-1 after birth in preterm infants arrests vascular growth. Relative hypoxia then increases the expression of VEGF. When IGF-1 expression gradually increases with the growth of preterm infants and exceeds the threshold under the condition of excessive VEGF, pathological angiogenesis occurs abruptly [242]. The interaction of VEGF and IGF-1 is summarized in Figure 1B.

Lofqvist et al. showed that IGF-1 binding protein-3 (IGFBP-3), which forms a complex with IGF-1 in the blood, reduced the avascular area and the rate of neovascularization by promoting vascular regrowth in OIR [243]. In preterm infants, the serum concentrations of IGF-1 and IGFBP-3 are known to fall rapidly below normal intrauterine levels [244]. Lofqvist et al. also revealed that the prolonged lower expression of IGFBP-3 is associated with ROP severity [243].

Based on the results of these basic experiments, it was hypothesized that elevated blood levels of IGF-1 and IGFBP-3 in the early postnatal period can inhibit the development and severity of ROP. However, in a randomized clinical trial, the intravenous administration of recombinant human IGF-1/IGFBP-3 had no significant effect on the incidence and severity of ROP [245].

### 4.6. Non-Steroidal Anti-Inflammatory Drugs

The expression of various inflammatory cytokines and chemokines is elevated in the vitreous fluid of ROP patients, and there is no doubt that inflammation is deeply involved in the pathogenesis of ROP [133,246]. There are various pathways involved inflammation-induced angiogenesis [247], and the arachidonic acid cascade is one of the most representative pathways [248]. Cyclooxygenases (COX) are enzymes that catalyze the biosynthesis of five bioactive prostanoids from arachidonic acid released from the cell membrane in response to tissue damage [249]. Prostanoids include prostaglandin (PG)D_2_, PGE_2_, PGF_2_, PGI_2_, and thromboxane A_2_, among which PGE_2_ is known to promote tumor angiogenesis [250]. Since PGE_2_ is synthesized by COX-2 [249], many researchers have investigated the roles of COX-2 and PGE_2_ in ROP pathogenesis. Nandgaonkar et al. showed that indomethacin, a non-steroidal anti-inflammatory drug (NSAID), had an inhibitory effect on angiogenesis in OIR [251]. Although indomethacin inhibits both COX-1 and COX-2, Sennlaub et al. reported that PGE_2_ synthesized by COX-2 caused retinal angiogenesis by binding to EP_3_ on vascular endothelial cells [252]. In addition, Yanni et al. demonstrated that the binding of PGE_2_ to EP_4_ increased VEGF production from Müller cells and caused pathological angiogenesis in OIR [253].

A clinical trial conducted to evaluate the therapeutic effect of indomethacin on patent ductus arteriosus revealed that indomethacin significantly reduced the incidence of ROP [254]. However, other studies yielded conflicting results [255,256,257]. Indeed, high doses of indomethacin were reported to increase the incidence of severe ROP [258]. Other studies using ibuprofen rather than indomethacin also found no significant effect of NSAIDs on the incidence of ROP [259,260,261]. Clinical trials using ophthalmic solutions containing NSAIDs (ketorolac tromethamine) have also been conducted. Some reported a reduction in the incidence of severe ROP with eye drop treatment [262], while others reported no effect [263]. Thus, the therapeutic efficacy of NSAID eye drops remains inconclusive.

### 4.7. ω-3-Polyunsaturated Fatty Acids

Docosahexaenoic acid and eicosapentaenoic acid are the major ω3-polyunsaturated fatty acids (ω3-PUFAs) in the retina and are present in the membranes of neural and vascular endothelial cells [264]. As with arachidonic acid, ω3-PUFAs are released as free fatty acids from the cell membrane by phospholipase A_2_ [265]. Connor et al. found that administering the ω3-PUFA diet to OIR mice reduced neovascularization [266]. They showed that the binding of ω3-PUFA-produced resolvin E1 to ChemR23, which is expressed in microglia, reduced tumor necrosis factor α (TNFα) production by microglia and consequently suppressed pathological angiogenesis (Figure 1) [266]. Sapieha showed that the catalysis of ω3-PUFA-produced 4-hydroxy-docosahexaenoic acid by 5-lipoxygenase suppressed endothelial cell proliferation via the activation of peroxisome proliferator-activated receptor γ (Figure 1B) [267].

Several clinical trials have assessed the effects of fish oil emulsion supplementation on ROP. Many trials have demonstrated the efficacy of ω3-PUFAs in this regard [268,269,270]. Najm et al. did not find an association between ω3-PUFAs and ROP incidence in their trial [271], but a secondary analysis revealed an association between the deficit of arachidonic acid, one of the ω6-PUFAs, and later ROP development [272].

### 4.8. β-Blockers

Rocci et al. hypothesized that myopic changes in preterm infants could be caused by an excess production of aqueous humor and examined the effect of the topical administration of timolol maleate on OIR [254,255]. Timolol eye drops not only reduced the intraocular pressure (IOP), but also inhibited vaso-obliteration and subsequent vaso-proliferation [273,274]. Increased blood perfusion because of reduced IOP was thought to the cause of the suppression of disease activity, but it was unclear why the β-blocker was effective [273,274].

Léauté-Labrèze et al. and Lutgendorf et al. reported the therapeutic effects of β-blockers on hemangiomas and malignancies, suggesting that they had an anti-angiogenic effect [275,276]. It was then shown that noradrenaline (NA) causes angiogenesis by increasing the expression of VEGF and matrix metalloproteinases in tumor cells and also acts on the vascular endothelial cells themselves to promote VEGF production [277,278]. In OIR, the adrenaline and NA/β-adrenergic receptor signaling pathway is also involved in pathological angiogenesis, while the administration of propranolol, a β-blocker, was found to significantly reduce neovascularization [279]. Ristori et al. reported that the administration of propranolol reduced VEGF expression in the mouse retina and suppressed vascular leakage via the inhibition of occludin redistribution [280]. These results suggested that β-blockers could reduce ROP progression by directly inhibiting pathological angiogenesis, rather than by ameliorating the retinal relative hypoxia (Figure 1).

Several studies evaluated the effects of oral administration of propranolol [281,282,283,284], and Ozturk et al. reported a significant reduction in ROP progression from stage 2 [281]. However, Filippi et al. reported that oral propranolol carries a risk of serious adverse events such as hypotension and bradycardia [285], leading them to assess the effects of propranolol eye drops [286,287]. This group demonstrated that propranolol 0.2% eye drops inhibited progression to stage ≥2/3 [287]. The trials using propranolol eye drops observed no serious adverse events, and therefore propranolol has the potential to become a new treatment for ROP.

### 4.9. Rho-Associated Protein Kinase

Rho-associated protein kinase (ROCK) is a downstream effector of the small guanosine-5′-triphosphate-binding protein, RhoA [288]. ROCK has been shown to be a candidate molecular target for cardiovascular disease, neurological disorders, and ocular disease [289,290,291]. ROCK also contributes to inflammation, fibrovascular membrane formation, and retinal pigment epithelial dysfunction in diabetic retinopathy [292,293,294]. We previously revealed that ROCK activity is enhanced in the mouse retina by hypoxia and VEGF stimulation [295,296]. We have also shown that eye drops containing ripasudil, a selective ROCK inhibitor, reduce the avascular area and pathological neovascularization in OIR [296]. ROCK acts as a downstream molecule of the VEGF signaling pathway and causes increased vascular permeability by disrupting the tight junctions between vascular endothelial cells [295]. ROCK also promotes the infiltration of leukocytes into the retina by altering the cytoskeleton and the activation of leukocytes in the retina [295,297]. We have reported that leukocytes infiltrate the OIR retina, and among these leukocytes, M2 macrophages are deeply involved in the initiation of pathological angiogenesis [298,299]. These results suggest that ripasudil may inhibit pathological angiogenesis mainly via the suppression of retinal inflammation.

We are currently conducting a clinical trial to evaluate the efficacy of ripasudil eye drops in patients with stage 1 ROP (NCT04621136). A clinical trial by another group using fasudil ophthalmic solution, another ROCK inhibitor, is also in progress (NCT04191954).

## 5. New Biomarkers and Candidate Targeted Molecules for ROP

### 5.1. Macrophage Inflammatory Protein-1β

In our previous study using exhaustive genetic analysis, we showed that macrophage inflammatory protein-1β (MIP-1β) expression was significantly increased in the retinas of OIR mice [300,301]. MIP-1β, also known as chemokine CC motif ligand 4, promotes the migration of leukocytes to inflamed tissues. Because MIP-1β is rapidly upregulated by relative hypoxic stimulation [300], its expression may be a highly sensitive indicator of retinal ischemia. Unfortunately, intravitreal injection of neutralizing antibodies against MIP-1β inhibited physiological angiogenesis and exacerbated pathological angiogenesis [301]. Therefore, MIP-1β may be a potentially useful biomarker of ROP progression and recurrence rather than a therapeutic target molecule.

### 5.2. Mast Cell Tryptase

It has been reported that mast cells cause tumor angiogenesis by releasing inflammatory cytokines and proteolytic enzymes such as tryptase and chymase [302,303]. Matsuda et al. demonstrated that mast cell tryptase (MCT) is essential for retinal pathological angiogenesis in OIR [304]. Transient receptor potential ankyrin 1 expressed on mast cell membranes, which acts as an oxygen sensor, triggers the degranulation of mast cells upon stimulation by relative hypoxia. They found that MCTs upregulated the expression of MCP-1 and VEGF in the OIR retina and acted on vascular endothelial cells directly to promote the production of MCP-1, an angiogenic chemokine. They also observed elevated MCT concentrations in the blood of patients with severe ROP, suggesting that MCT may become a novel biomarker and therapeutic target molecule in the future.

### 5.3. Suppressor of Cytokine Signaling-3 and Retinoic Acid Receptor-Related Orphan Receptor Alpha

Suppressor of cytokine signaling-3 (SOCS3) acts as an endogenous angiogenesis inhibitor in vivo [305]. SOCS3, the expression of which is induced by pro-inflammatory cytokines such as TNFα and interleukin-6, is also known to act as a negative feedback regulator of inflammation and growth factor signaling [306,307]. Stahl et al. found that reductions in SOCS3 expression in endothelial cells increased neovascularization in OIR [306]. They revealed that TNFα-induced SOCS3 expression suppressed endothelial cell proliferation by inhibiting the IGF-1/mammalian target of the rapamycin pathway [306]. Sun et al. confirmed that SOCS3 inhibits the signal transducer and activator of transcription 3/VEGF pathway and that SOCS3 reduction in neurons and glial cells promotes pathological angiogenesis in the OIR retina [308]. Moreover, Sun et al. showed that retinoic acid receptor-related orphan receptor α (RORα) stimulates inflammatory cytokine production and suppresses SOCS3 transcription in retinal macrophages and microglia [309]. Increased SOCS3 expression by RORα inhibition suppressed retinal neovascular formation in OIR [309]. These results suggest that the inhibition of RORα or induction of SOCS3 may be a novel treatment for ROP.

## 6. Clinical Trials in Preterm Infants: Differences from Adults

Before conducting a clinical trial in children with a new drug (compound or antibody product), its safety must be fully validated in non-clinical studies. Even existing drugs that are known to be safe in adults are not guaranteed to be harmless in preterm infants. Pharmacokinetics, such as the absorption, distribution, metabolism, and excretion of drugs, can vary significantly with patient age and may result in differences in efficacy and toxicity. Additional safety and toxicity studies using young animals should be conducted as needed, particularly because almost all organs in preterm infants are immature.

In recent years, there has been an increase in the number of clinical trials evaluating the efficacy of eye drops for ROP. Because preterm infants are known to produce less tear fluid than full-term infants [310], the increased concentration of drugs in the lacrimal sac should also be considered. In addition, drug-induced contact dermatitis may be more severe owing to immature epithelial barrier function [311]. Preterm infants are at risk of unexpected and severe adverse events. A classification of the severity of neonatal adverse events has been proposed by the International Neonatal Consortium [312].

## 7. Future Prospects

With the advent of various comprehensive analysis methods (single-cell ribonucleic acid-sequence, proteome analysis, etc.) in recent years, the molecular mechanisms involved in the development and severity of ROP are likely to be further elucidated in detail. Analysis of OIR retinas and patient samples such as serum or aqueous humor will reveal new therapeutic target molecules.

## 8. Conclusions

In this review, we presented the pathophysiology of ROP, existing treatments, clinical trials, and candidate molecules expected to become novel therapeutic targets for ROP. In the last 80 years since ROP was defined, a variety of treatments have been developed. OIR, an animal model of ROP, has been developed based on clinical experience, and many therapeutic modalities have been developed from basic research using OIR. In brief, it can be stated that the foundation for translational research has been firmly established in the field of ROP. Safer and less invasive treatments are expected to emerge from basic research in the future.

## Figures and Tables

**Figure 1 jcm-10-00331-f001:**
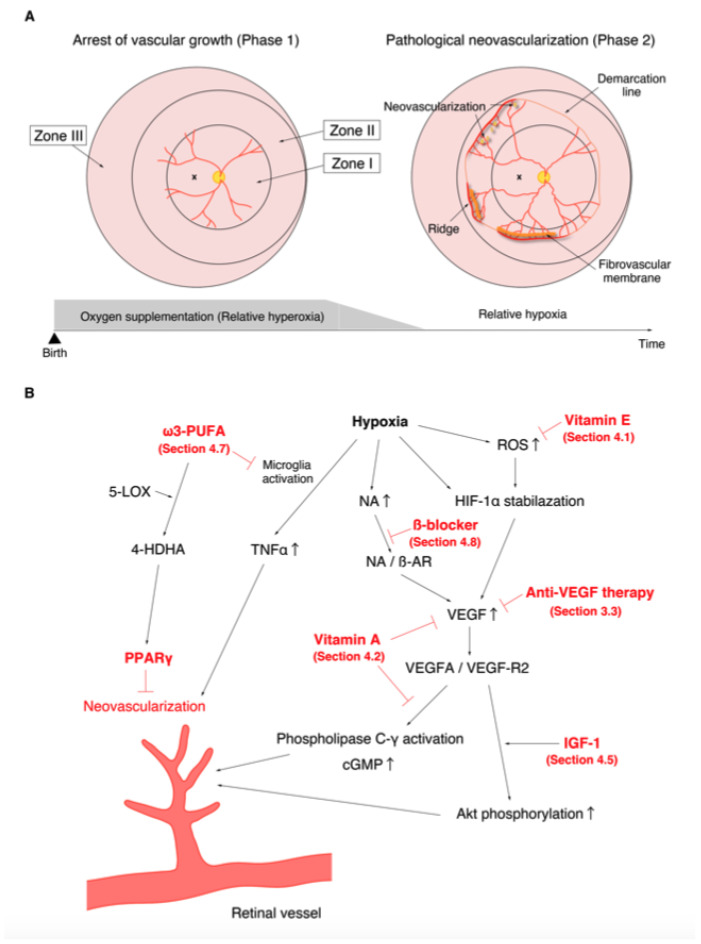
Phases of retinopathy of prematurity (ROP) and the action mechanisms of drugs with confirmed therapeutic effects in the treatment of ROP in previous clinical trials and meta-analysis. (**A**). Relative hyperoxia in phase 1 arrests vascular growth (left). Relative hypoxia in phase 2 causes pathological neovascularization (right). (**B**). Molecular mechanisms of neovascularization. The various points at which anti-vascular endothelial growth factor therapy (Section 3.3), vitamin E (Section 4.1), vitamin A (Section 4.2), insulin-like growth factor-1 (Section 4.5), ω3-polyunsaturated fatty acids (Section 4.7), and β-blockers (Section 4.8) act are shown in red.

**Figure 2 jcm-10-00331-f002:**
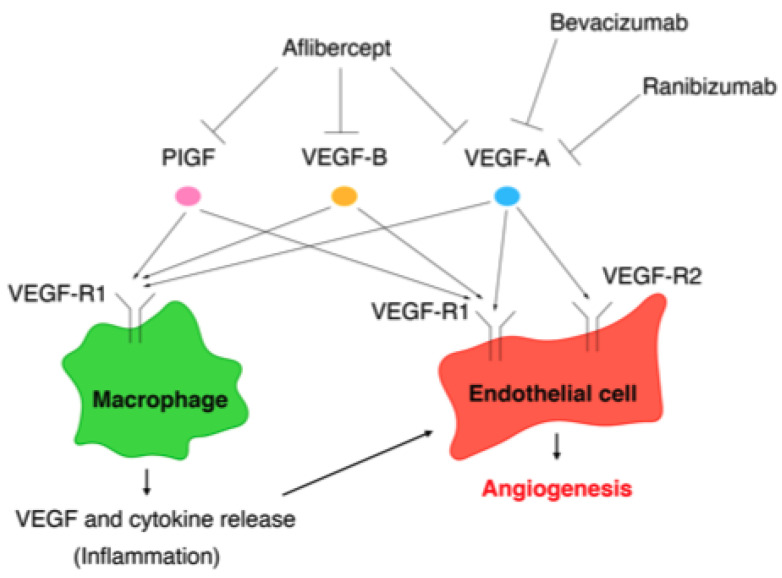
Vascular endothelial growth factor/vascular endothelial growth factor (VEGF) receptor pathways involved in angiogenesis and the targets of VEGF (anti-VEGF agents): bevacizumab, ranibizumab, and aflibercept.

**Table 1 jcm-10-00331-t001:** The classification of retinopathy of prematurity (ROP).

Location (see Figure 1)	Zone I	Circle centered on the optic disc with the radius of twice the distance from the optic disc to the fovea.
	Zone II	Circle centered on the optic disc with the distance to the nasal ora serrata as the radius, excluding Zone I
	Zone III	Residual retina anterior to Zone II.
Severity	Stage 1	Demarcation line
	Stage 2	Ridge
	Stage 3	Ridge with extra retinal fibrovascular proliferation
	Stage 4	Subtotal retinal detachment
	Stage 5	Total retinal detachment
Plus disease		Venous dilation and arterial tortuosity of the posterior pole vessels

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
