# Peer review of "Translational Research in Retinopathy of Prematurity: From Bedside to Bench and Back Again"

_jcm, 2021, doi:10.3390/jcm10020331_

Round 1
Reviewer 1 Report
In this manuscript, Arima et.al, reviews how basic Retinopathy of Prematurity (ROP) research has evolved with clinical experience and the subsequent emergence of new drugs. This review is a valuable addition to the field as this review summarizes the existing literature on pathophysiology of ROP, existing treatments, trials and presents the candidate molecules expected to become novel targets for ROP. The manuscript is well written, and figures are well presented. The strength of the review is its capacity to offer a complete picture of the topic ranging from the clinical aspects of ROP, to the different molecular mechanisms of neovascularization in ROP, along with clinical trials, and new biomarkers and candidate targeted molecules for ROP. The article has interesting observations which are beneficial to researchers in the areas of ROP and other retinal associated complications. The reviewer does not notice significant and general limitations of the manuscript apart from the minor concerns discussed below.
- Diagram/table or further description about stages of ROP will be beneficial to the readers as the stages are mentioned few times in the manuscript.
-Line 103 : Clarify abbreviation of SpO2
-Line 131 : Clarify abbreviation of PaO2
-Line 160 to 162 : Add more details from ETROP trial
-Line 362 : Clarify abbreviation of IκB
-Conclusion - Discuss further on comprehensive analysis methods which could advance ROP treatment.
Author Response
A point-by-point response to the issues raised by the reviewers
(Manuscript number: jcm-1066550)
We would like to thank the reviewers for insightful comments which have helped us to improve the manuscript. A point-by-point response to the issues raised is as follows.
Response to Reviewer 1
Comment : Diagram/table or further description about stages of ROP will be beneficial to the readers as the stages are mentioned few times in the manuscript.
We thank the reviewer’s comment. As the reviewer pointed out, it is unkind to the readers that the ROP classification is not explained even though we mentioned to ROP stage repeatedly. Therefore, we have summarized ROP classification in Table 1 (line 89 and 99).
Comment : Line 160 to 162 : Add more details from ETROP trial
We appreciate the reviewer’s comment. The ETROP trial not only proved the efficacy of the laser photocoagulation, but also established the criteria for the timing of therapeutic intervention. We noted in lines 184-186 that the ETROP trial gave rise to the concept of Type 1 ROP.
Comment :
-Line 103 : Clarify abbreviation of SpO2
-Line 131 : Clarify abbreviation of PaO2
-Line 362 : Clarify abbreviation of IκB
We thank the reviewer’s comment. We have added the official names of SpO2, PaO2 and IκB in line 112, 141 and 377, respectively.
Comment : -Conclusion - Discuss further on comprehensive analysis methods which could advance ROP treatment.
We appreciate the reviewer’s comment. Recent advances in comprehensive analysis methods have made it possible to easily identify molecular networks involved in the pathogenesis of diseases. We have described in lines 585-589 that the analysis of OIR retinas and patient samples is likely to reveal new candidate therapeutic target molecules.

Reviewer 2 Report
This is a well written review article compiling the literature on Retinopathy of Prematurity (ROP), a major cause of childhood vision problem. The review has covered on the pathogenesis, experimental models, and current and future treatment strategies. However a major missing element is the neuronal dysfunction associated with ROP. Even though the disease had been initially identified as a vascular disease, there are increasing evidences available in literature about visual dysfunction resulting from neuronal damage (significant research studies have been done by AB Fulton’s group).
The review will be significantly improved by the addition of the following.
- Introduction and pathogenesis sections need to be modified by including changes in neuronal dysfunction in ROP patients.
- Animal model section is underdeveloped and can be expanded by including the experimental models of constant hyperoxia and intermittent hyperoxia in rats and mouse (studies by Smith LE, Penn JS, Caldwell RB, Narayanan SP, Sennlaub F etc ).
- It is suggested to include inflammation/inflammatory signaling under ROP pathogenesis as it’s a major focus of many clinical trials.
- Readers will significantly benefit if the information on clinical trials provided under section 4 can be summarized in a table format (agents, target and inference).
- Please include a section stating the future prospects.
Author Response
A point-by-point response to the issues raised by the reviewers
(Manuscript number: jcm-1066550)
We would like to thank the reviewers for insightful comments which have helped us to improve the manuscript. A point-by-point response to the issues raised is as follows.
Response to Reviewer 2
Comment : Introduction and pathogenesis sections need to be modified by including changes in neuronal dysfunction in ROP patients.
We thank the reviewer’s comment. As the reviewer pointed out, changes in oxygen concentration not only promotes angiogenesis via hypoxia-induced signaling but also damages the photoreceptors directly. Visual dysfunction due to ROP is not caused by angiogenesis alone. We have described these contents in lines 46-50 and lines 143-145.
Comment : Animal model section is underdeveloped and can be expanded by including the experimental models of constant hyperoxia and intermittent hyperoxia in rats and mouse (studies by Smith LE, Penn JS, Caldwell RB, Narayanan SP, Sennlaub F etc ).
We thank the reviewer’s comment. We noted in lines 151-152 that there are various ways to load supplemental oxygen.
Comment : It is suggested to include inflammation/inflammatory signaling under ROP pathogenesis as it’s a major focus of many clinical trials.
We agree with the reviewer’s comment. Since inflammation is definitely involved in the onset and severity of ROP, we described in detail what inflammatory signals are involved in the pathogenesis of ROP in the section of “Translational ROP research” and “New biomarkers and candidate targeted molecules for ROP”.
Comment : Readers will significantly benefit if the information on clinical trials provided under section 4 can be summarized in a table format (agents, target and inference).
We appreciate the reviewer’s comment. Since ROP is a rare disease, it is difficult to conduct international clinical trials. In addition, the background of patients admitted in each NICU is very different. Therefore, even if the same drug is used, the results often differ from trial to trial. The figure quoted from the paper of meta-analysis is shown below (As images could not be inserted on the web, PDF file will be sent separately). The efficacy of vit A and vit E varies from trial to trial, but meta-analysis indicates that they can prevent the onset or progression of ROP. Because this meta-analysis was conducted using the random effect model, vit A and vit E are likely to be effective therapeutic agents for ROP. As the reviewer pointed out, we would like to include the results of each trial if possible, but we think there is a risk of confusing the readers. We thus changed the description of Figure 1B from “in previous clinical trials” to "in previous clinical trials and meta-analysis".
Figure only to the Reviewer 2 (quoted from Ref. 4): The meta-analysis showed the therapeutic efficacy of vit A and vit E, but the relative risk is different in each study.
Comment : Please include a section stating the future prospects.
We thank the reviewer’s comment. According to the reviewer's recommendation, we have added a section describing the future prospects (lines 585-589).

Round 2
Reviewer 2 Report
In the revised manuscript, the authors have not adequately responded to the comments provided by the reviewer. They have only vaguely addressed most of them.
- Authors need to review the available literature and include multiple references demonstrating neuronal damage dysfunction in ROP. Photoreceptors are only one type of neurons affected.
- Changes in the animal model section are done very poorly just by adding one sentence! This section needs to be expanded in detail. Plenty of literature is available on these models.
- Inflammatory changes are not included under ROP pathogenesis.
Author Response
A point-by-point response to the issues raised by the reviewer 2
(Manuscript number: jcm-1066550)
We would like to thank the reviewer 2 for insightful comments which have helped us to improve the manuscript. A point-by-point response to the issues raised is as follows.
Response to Reviewer 2
Comment : Authors need to review the available literature and include multiple references demonstrating neuronal damage dysfunction in ROP. Photoreceptors are only one type of neurons affected.
We thank the reviewer’s comment. As the reviewer pointed out, it is not only photoreceptor cells that are damaged by ROP. ROP can damage all the neurons (bipolar cells, amacrine cells and ganglion cells) that consist of the neural retina. Therefore, we have added the literature on human ROP which investigated neural damage using retinal imaging and ERG to Section 1 (lines 46-50) and the literature on OIR to Section 2.3 (line190-194).
In addition, the following references have been newly cited.
[References added to Section 1]
1.  Provis, J.M. Development of the primate retinal vasculature. Prog. Retin. Eye Res. 2001, 20, 799–821.
- Dubis, A.M.; Costakos, D.M.; Subramaniam, C.D.; Godara, P.; Wirostko, W.J.; Carroll, J.; Provis, J.M. Evaluation of normal human foveal development using optical coherence tomography and histologic examination. Arch. Ophthalmol. 2012, 130, 1291–1300.
- Ecsedy, M.; Szamosi, A.; Karko, C.; Zubovics, L.; Varsanyi, B.; Nemeth, J.; Recsan, Z. A comparison of macular structure imaged by optical coherence tomog- raphy in preterm and full-term children. Invest. Ophthalmol. Vis. Sci. 2007, 48, 5207–5211.
- Hammer, D.X.; Iftimia, N.V.; Ferguson, R.D.; Bigelow, C.E.; Ustun, T.E.; Barnaby, A.M.; Fulton, A.B. Foveal fine structure in retinopathy of prematurity: an adaptive optics Fourier domain optical coherence tomography study. Invest. Ophthalmol. Vis. Sci. 2008, 49, 2061–2070.
- Fulton, A.B.; Hansen, R.M.; Moskowitz, A. The cone electroretinogram in retinopathy of prematurity. Invest. Ophthalmol. Vis. Sci. 2008, 49, 814–819.
- Miki, A.; Yamada, Y.; Nakamura, M. The size of the foveal avascular zone is associated with foveal thickness and structure in premature children. J Ophthalmol. 2019, 2019, 8340729.
[References added to Section 2.3]
- Wellard, J.; Lee, D.; Valter, K.; Stone, J. Photoreceptors in the rat retina are specifically vulnerable to both hypoxia and hyperoxia. Vis. Neurosci. 2005, 22, 501–507.
- Fulton, A.B.; Reynaud, X.; Hansen, R.M.; Lemere, C.A.; Parker, C.; Williams, T.P. Rod photoreceptors in infant rats with a history of oxygen exposure. Invest. Ophthalmol. Vis. Sci. 1999, 40, 168–174.
- Mezu-Ndubuisi, O.J.; Macke, E.L.; Kalavacherla, R; Nwaba, A.A.; Suscha, A.; Zaitoun, I.S.; Ikeda, A.; Sheibani, N. Long-term evaluation of retinal morphology and function in a mouse model of oxygen-induced retinopathy. Mol Vis. 2020, 26, 257–276.
Comment : Changes in the animal model section are done very poorly just by adding one sentence! This section needs to be expanded in detail. Plenty of literature is available on these models.
We appreciate the reviewer’s comment. In Section 2.3, we have described in what animal species has OIR been created, and how improvements have been made to the oxygen exposure method to efficiently induce retinal neovascularization (lines 232-256).
The following references have been newly cited.
- Kremer, I.; Kissun, R.; Nissenkorn, I.; Ben-Sira, I.; Garner, A. Oxygen-induced retinopathy in newborn kittens. A model for ischemic vasoproliferative retinopathy. Invest. Ophthalmol. Vis. Sci. 1987, 28, 126–130.
- Phelps, D.L. Reduced severity of oxygen-induced retinopathy in kittens recovered in 28% oxygen. Pediatr. Res. 1988, 24, 106–109.
- Flower, R.W.; Blake, D.A. Retrolental fibroplasia: evidence for a role of the prostaglandin cascade in the pathogenesis of oxygen-induced retinopathy in the newborn beagle. Pediatr. Res. 1981, 15, 1293–1302.
- McLeod, D.S.; Brownstein, R.; Lutty, G.A. Vaso-obliteration in the canine model of oxygen-induced retinopathy. Invest. Ophthalmol. Vis. Sci. 1996, 37, 300–311.
- McLeod, D.S.; Crone, S.N.; Lutty, G.A. Vasoproliferation in the neonatal dog model of oxygen-induced retinopathy. Invest. Ophthalmol. Vis. Sci. 1996, 37, 1322–1333.
- McLeod, D.S.; D'Anna, S.A.; Lutty, G.A. Clinical and histopathologic features of canine oxygen-induced proliferative retinopathy. Invest. Ophthalmol. Vis. Sci. 1998, 39, 1918–1932.
- Patz, A. Oxygen studies in retrolental fibroplasia. IV. Clinical and experimental observations. Am. J Ophthalmol. 1954, 38, 291–308.
- Ashton, N.; Blach, R. Studies on developing retinal vessels. VIII. Effect of oxygen on the retinal vessels of the ratling. Br. J Ophthalmol. 1961, 45, 321–340.
- Penn, J.S.; Tolman, B.L.; Lowery, L.A. Variable oxygen exposure causes preretinal neovascularization in the newborn rat. Invest. Ophthalmol. Vis. Sci. 1993, 34, 576–585.
- Penn, J.S.; Henry, M.M.; Tolman, B.L. Exposure to alternating hypoxia and hyperoxia causes severe proliferative retinopathy in the newborn rat. Pediatr. Res. 1994, 36, 724–731.
- Kubota, Y.; Takubo, K.; Shimizu, T.; Ohno, H.; Kishi, K.; Shibuya, M.; Saya, H.; Suda, T. M-CSF inhibition selectively targets pathological angiogenesis and lymphangiogenesis. J Exp. Med. 2009, 206, 1089–1102.
Comment : Inflammatory changes are not included under ROP pathogenesis.
We thank the reviewer’s comment. We thought the title of section 2, "Pathophysiology of ROP," was inappropriate, as it merely mentions that fluctuations in oxygen concentration promote retinal angiogenesis in preterm infants. The action mechanisms of the candidate target molecules identified by the analysis of the OIR retinas are described in more detail in Sections 3-5 than in Section 2. Therefore, the title of Section 2 has been changed from "Pathophysiology of ROP" to "Relationship between oxygen concentration and retinal neovascularization. In addition, at the end of Section 2, we added a statement that we intend to introduce how the candidate target molecules are thought to be involved in the development and severity of ROP, i.e., the pathophysiology of ROP will be detailed in Section 3 and beyond (lines 279-288).
